# Unsupervised Word Discovery with Segmental Neural Language Models

## Abstract

We propose a segmental neural language model that combines the representational power of neural networks and the structure learning mechanism of Bayesian non-parametrics, and show that it learns to discover semantically meaningful units (e.g., morphemes and words) from unsegmented character sequences. The model generates text as a sequence of segments, where each segment is generated either character-by-character from a sequence model or as a single draw from a lexical memory that stores multi-character units. Its parameters are fit to maximize the marginal likelihood of the training data, summing over all segmentations of the input, and its hyperparameters are likewise set to optimize held-out marginal likelihood. To prevent the model from overusing the lexical memory, which leads to poor generalization and bad segmentation, we introduce a differentiable regularizer that penalizes based on the expected length of each segment. To our knowledge, this is the first demonstration of neural networks that have predictive distributions better than LSTM language models and also infer a segmentation into word-like units that are competitive with the best existing word discovery models.

## 1 Introduction

How infants discover the words of their native languages is a long-standing question in developmental psychology (Saffran et al., 1996). Machine learning has contributed much to this discussion by showing that predictive models of language are capable of inferring the existence of word boundaries solely based on statistical properties of the input (Elman, 1990; Brent & Cartwright, 1996; Goldwater et al., 2009). Unfortunately, the best language models, measured in terms of their ability to model language, segment quite poorly (Chung et al., 2017; Wang et al., 2017), while the strongest models in terms of word segmentation are far too weak to adequately predict language (Goldwater et al., 2009; Berg-Kirkpatrick et al., 2010). Moreover, since language acquisition is ultimately a multimodal process, neural models which simplify working with multimodal data offer opportunities for future research. However, as Kádár et al. (2018) have argued, current neural models' inability to discover meaningful words is too far behind the current (non-neural) state-of-the-art to be a useful foundation.

In this paper, we close this gap by introducing a neural model (§2) of natural language sentences that explicitly discovers and models word-like units from completely unsegmented sequences of characters. The model generates text as a sequence of segments, where each segment is generated either character-by-character from a sequence model or as a single draw from a lexical memory of multi-character units. The segmentation decisions and decisions about the generation mechanism for each segment are latent. In order to efficiently deal with an exponential number of possible segmentations, we use a conditional semi-Markov model. The the characters inside each segment are generated using non-Markovian processes, conditional on the previously generated characters (the previous segmentation decisions are forgotten). This conditional independence assumption—forgetting previous segmentation decisions—enables us to calculate and differentiate exact marginal likelihood over all possible discrete segmentation decisions with a dynamic programming algorithm, while letting the model retain the most relevant information about the generation history.

There are two components to make the model work. One is a lexical memory. The memory stores pairs of a vector (key) and a string (value) appearing in the training set and the vector representation of each strings are randomly initialized and learned during training. The other is a regularizer (§3) to prevent the model from overfitting to the training data. Since the lexical memory stores strings

that appeared in the training data, each sentence could be generated as a single unit, thus the model can fit to the training data perfectly while generalizing poorly. The regularizer penalizes based on the expectation of the powered length of each segment. Although the length of each segment is not differentiable, the expectation is differentiable and can be computed efficiently together with the marginal likelihood for each sentence in a single forward pass.

Our evaluation (§4–§6), therefore, looks at both language modeling performance and the quality of the induced segmentations. First, we look at the segmentations induced by our model. We find that these correspond closely to human intuitions about word segments, competitive with the best existing models. These segments are obtained in models whose hyperparameters are tuned to optimize validation likelihood, whereas tuning the hyperparameters based on likelihood on our benchmark models produces poor segmentations. Second, we confirm findings (Kawakami et al., 2017; Mielke & Eisner, 2018) that show that word segmentation information leads to better language models compared to pure character models. However, in contrast to previous work, we do so without observing the segment boundaries, including in Chinese, where word boundaries are not part of the orthography. Finally, we find that both the lexicon and the regularizer are crucial for good performance, particularly in word segmentation—removing either or both significantly harms performance.

## 2 MODEL

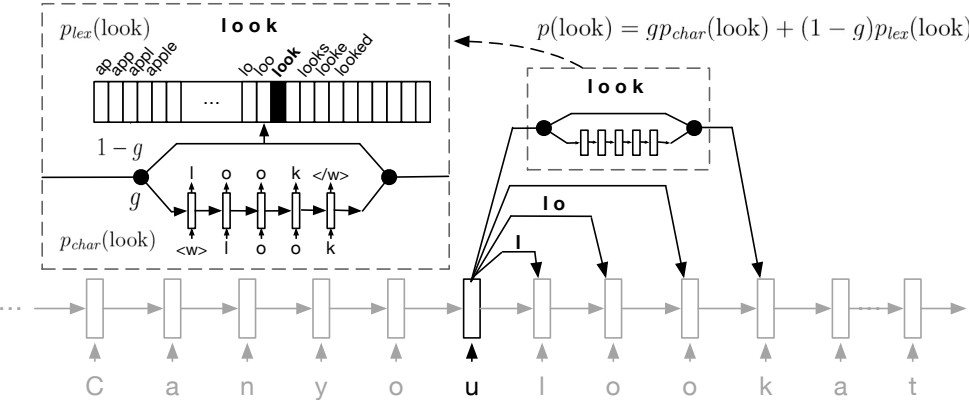

Figure 1: Fragment of the segmental neural language model, as it is used to evaluate the marginal likelihood of a sequence. At the indicated time, the model has previously generated the sequence *Canyou*, and four possible continuations are shown.

We now describe the segmental neural language model (SNLM). Refer to Figure 1 for an illustration. The SNLM generates a character sequence $x = x_1, \ldots, x_n$, where each $x_i$ is a character in a finite character set $\Sigma$. Each sequence $x$ is the concatenation of a sequence of segments $\underline{s} = s_1, \ldots, s_{|\underline{s}|}$ where $|\underline{s}| \leq n$ measures the length of the sequence in segments and each segment $s_i \in \Sigma^+$ is a sequence of characters, $s_{i,1}, \ldots, s_{i,|s_i|}$. Intuitively, each $s_i$ corresponds to one word. Let $\pi(s_1, \ldots, s_i)$ represent the concatenation of the characters of the segments $s_1$ to $s_i$, discarding segmentation information; thus $x = \pi(\underline{s})$. For example if $x = \texttt{anapple}$, the underlying segmentation might be $\underline{s} = \texttt{an apple}$ (with $s_1 = \texttt{an}$ and $s_2 = \texttt{apple}$), or $\underline{s} = \texttt{a nap ple}$, or any of the $2^{|x|-1}$ segmentation possibilities for $x$.

The SNLM defines the distribution over $x$ as the marginal distribution over all segmentations that give rise to $x$, i.e.,

$$p(x) = \sum_{\underline{s}:\pi(\underline{s})=x} p(\underline{s}). \tag{1}$$

To define the probability of $p(\underline{s})$, we use the chain rule, rewriting this in terms of a product of the series of conditional probabilities, $p(s_t \mid \underline{s}_{<t})$. The process stops when a special end-sequence segment $\langle/\text{s}\rangle$ is generated. To ensure that the summation in Eq. 1 is tractable, we assume the

following:

$$p(\boldsymbol{s}_t \mid \underline{\boldsymbol{s}}_{<t}) \approx p(\boldsymbol{s}_t \mid \pi(\underline{\boldsymbol{s}}_{<t})) = p(\boldsymbol{s}_t \mid \boldsymbol{x}_{<t}), \tag{2}$$

which amounts to a conditional semi-Markov assumption—i.e., non-Markovian generation happens inside each segment, but the segment generation probability does not depend on memory of the previous segmentation decisions, only upon the sequence of characters $\pi(\underline{\boldsymbol{s}}_{<t})$ corresponding to the prefix character sequence $\boldsymbol{x}_{<t}$. This assumption has been employed in a number of related models to permit the use of LSTMs to represent rich history while retaining the convenience of dynamic programming inference algorithms (Wang et al., 2017; Ling et al., 2017; Graves, 2012).

## 2.1 Segment generation

We model $p(\boldsymbol{s}_t \mid \boldsymbol{x}_{<t})$ as a mixture of two models, one that generates the segment using a sequence model and the other that generates multi-character sequences as a single event. Both are conditional on a common representation of the history, as is the mixture proportion.

**Representing history**  To represent $\boldsymbol{x}_{<t}$, we use an LSTM encoder to read the sequence of characters, where each character type $\sigma \in \Sigma$ has a learned vector embedding $\mathbf{v}_\sigma$. Thus the history representation at time $t$ is $\mathbf{h}_t = \mathrm{LSTM}_{enc}(\mathbf{v}_{x_1}, \ldots, \mathbf{v}_{x_t})$. This corresponds to the standard history representation for a character-level language model, although in general we assume that our modeled data is not delimitered by whitespace.

**Character-by-character generation**  The first component model, $p_{char}(\boldsymbol{s}_t \mid \mathbf{h}_t)$, generates $\boldsymbol{s}_t$ by sampling a sequence of characters from a LSTM language model over $\Sigma$ and a two extra special symbols, an end-of-word symbol $\langle /\mathrm{w} \rangle \notin \Sigma$ and the end-of-sequence symbol $\langle /\mathrm{s} \rangle$ discussed above. The initial state of the LSTM is a learned transformation of $\mathbf{h}_t$, the initial cell is $\mathbf{0}$, and different parameters than the history encoding LSTM are used. During generation, each letter that is sampled (i.e., each $s_{t,i}$) is fed back into the LSTM in the usual way and the probability of the character sequence decomposes according to the chain rule. The end-of-sequence symbol can never be generated in the initial position.

**Lexical generation**  The second component model, $p_{lex}(\boldsymbol{s}_t \mid \mathbf{h}_t)$, samples full segments from lexical memory. Lexical memory is a key-value memory containing $M$ entries, where each key, $\mathbf{k}_i$, a vector, is associated with a value $\boldsymbol{v}_i \in \Sigma^+$. The generation probability of $\boldsymbol{s}_t$ is defined as

$$\mathbf{h}'_t = \mathrm{MLP}(\mathbf{h}_t)$$
$$\mathbf{m} = \mathrm{softmax}(\mathbf{K}\mathbf{h}'_t + \mathbf{b})$$
$$p_{lex}(\boldsymbol{s}_t \mid \mathbf{h}_t) = \sum_{i=1}^{M} m_i[\boldsymbol{v}_i = \boldsymbol{s}_t],$$

where $[\boldsymbol{v}_i = \boldsymbol{s}_t]$ is 1 if the $i$th value in memory is $\boldsymbol{s}_t$ and 0 otherwise, and $\mathbf{K}$ is a matrix obtained by stacking the $\mathbf{k}_i^\top$'s. Note that this generation process will assign zero probability to most strings, but the alternate character model can generate anything in $\Sigma^+$.

In this work, we fix the $\boldsymbol{v}_i$'s to be subsequences of at least length 2, and up to a maximum length $L$ that are observed at least $F$ times in the training data. These values are tuned as hyperparameters (See Appendix B for details of the reported experiments).

**Mixture proportion**  The mixture proportion, $g_t$, determines how likely the character generator is to be used at time $t$ (the lexicon is used with probability $1 - g_t$). It is defined by as $g_t = \sigma(\mathrm{MLP}(\mathbf{h}_t))$.

**Total segment probability**  The total generation probability of $\boldsymbol{s}_t$ is thus:

$$p(\boldsymbol{s}_t \mid \boldsymbol{x}_{<t}) = g_t p_{char}(\boldsymbol{s}_t \mid \mathbf{h}_t) + (1 - g_t) p_{lex}(\boldsymbol{s}_t \mid \mathbf{h}_t).$$

## 2.2 Inference

We are interested in two inference questions: first, given a sequence $\boldsymbol{x}$, evaluate its (log) marginal likelihood; second, given $\boldsymbol{x}$, find the most likely decomposition into segments $\underline{\boldsymbol{s}}^*$.

**Marginal likelihood** To efficiently compute the marginal likelihood, we use a variant of the forward algorithm for semi-Markov models (Yu, 2010), which incrementally computes a sequence of probabilities, $\alpha_i$, where $\alpha_i$ is the marginal likelihood of generating $\boldsymbol{x}_{\leq i}$ and concluding a segment at time $i$. Although there are an exponential number of segmental decompositions of $\boldsymbol{x}$, these values can be computed using $O(|\boldsymbol{x}|)$ space and $O(|\boldsymbol{x}|^2)$ time as:

$$\alpha_0 = 1, \qquad \alpha_t = \sum_{j=t-L}^{t-1} \alpha_j p(\boldsymbol{s} = \boldsymbol{x}_{j:t} \mid \boldsymbol{x}_{<j}). \tag{3}$$

By letting $x_{t+1} = \langle /\textsc{s} \rangle$, then $p(\boldsymbol{x}) = \alpha_{t+1}$.

**Most probable segmentation** The most probable segmentation of a sequence $\boldsymbol{x}$ can be computed by replacing the summation with a $\max$ operator in Eq. 3 and maintaining backpointers.

## 3 EXPECTED LENGTH REGULARIZATION

When the lexical memory contains all the substrings in the training data, the model easily overfits by copying the longest continuation from the memory. To prevent overfitting, we introduce a regularizer that penalizes based on the expectation of the exponentiated (by a hyperparameter $\beta$) length of each segment:

$$R(\boldsymbol{x}, \beta) = \sum_{\underline{\boldsymbol{s}}:\pi(\underline{\boldsymbol{s}})=\boldsymbol{x}} p(\underline{\boldsymbol{s}} \mid \boldsymbol{x}) \sum_{\boldsymbol{s} \in \underline{\boldsymbol{s}}} |\boldsymbol{s}|^\beta.$$

This can be understood as a regularizer based on the double exponential prior identified to be effective in previous work (Liang & Klein, 2009; Berg-Kirkpatrick et al., 2010). This expectation is a differentiable function of the model parameters. Because of the linearity of the penalty across segments, it can be computed efficiently using the above dynamic programming algorithm under the expectation semiring (Eisner, 2002). This is particular efficient since the expectation semiring jointly computes the expectation and marginal likelihood.

### 3.1 TRAINING OBJECTIVE

The model parameters are trained by minimizing the penalized log likelihood of a training corpus $\mathcal{D}$ of unsegmented sentences,

$$\mathcal{L} = \sum_{\boldsymbol{x} \in \mathcal{D}} [-\log p(\boldsymbol{x}) + \lambda R(\boldsymbol{x}, \beta)].$$

## 4 DATASETS

We evaluate our model on both English and Chinese segmentation. For both languages we used standard datasets for word segmentation and language modeling. For all datasets, we used train, validation and test splits.[1] Since our model assumes a closed character set, we removed validation and test samples which contain characters that do not appear in the training set. In the English corpora, whitespace characters are removed. In Chinese, they are not present to begin with. Refer to Appendix A for dataset statistics.

### 4.1 ENGLISH

**Brent Corpus** The Brent corpus is a standard corpus used in statistical modeling of child language acquisition (Brent, 1999; Venkataraman, 2001).[2] The corpus contains transcriptions of utterances directed at 13- to 23-month-old children. The corpus has two variants: an orthographic one (**BR-text**) and a phonemic one (**BR-phono**), where each character corresponds to a single English phoneme. As the Brent corpus does not have a standard train and test split, and we want to tune the parameters by measuring the fit to held-out data, we used the first 80% of the utterances for training and the next 10% for validation and the rest for test.

---

[1]The data and splits used in these experiments are available from `anonymous`.
[2]`https://childes.talkbank.org/derived`

**English Penn Treebank (PTB)**   We use the commonly used version of the PTB prepared by Mikolov et al. (2010). However, since we removed space symbols from the corpus, our cross entropy results cannot be compared to those usually reported on this dataset.

## 4.2 CHINESE

Since Chinese orthography does not mark spaces between words, there have been a number of efforts to annotate word boundaries. We evaluate against two corpora that have been manually segmented according different segmentation standards.

**Beijing University Corpus (PKU)**   The Beijing University Corpus was one of the corpora used for the International Chinese Word Segmentation Bakeoff (Emerson, 2005).

**Chinese Penn Treebank (CTB)**   We use the Penn Chinese Treebank Version 5.1 (Xue et al., 2005). It generally has a coarser segmentation than PKU (e.g., in CTB a full name, consisting of a given name and family name, is a single token), and it is a larger corpus.

## 5 EXPERIMENTS

We compare our model to benchmark Bayesian models, which are currently the best known unsupervised word discovery models, as well as to a simple deterministic segmentation criterion based on surprisal peaks (Elman, 1990) on language modeling and segmentation performance. Although the Bayeisan models are shown to able to discover plausible word-like units, we found that a set of hyper-parameters that provides best performance with such model on language modeling does not produce good structures as reported in previous works. This is problematic since there is no objective criteria to find hyper-parameters in fully unsupervised manner when the model is applied to completely unknown languages or domains. Thus, our experiments are designed to assess how well the models infers word segmentations of unsegmented inputs when they are trained and tuned to maximize the likelihood of the held-out text.

**DP/HDP Benchmarks**   Among the most effective existing word segmentation models are those based on hierarchical Dirichlet process (HDP) models (Goldwater et al., 2009; Teh et al., 2006). These generate a corpus of utterances segment by segment using the following process:

$$\theta_{\cdot} \sim \mathrm{DP}(\alpha_0, p_0)$$
$$\theta_{\cdot|\boldsymbol{s}} \sim \mathrm{DP}(\alpha_1, \theta_{\cdot}) \qquad \forall \boldsymbol{s} \in \Sigma^*$$
$$\boldsymbol{s}_{t+1} \mid \boldsymbol{s}_t \sim \mathrm{Categorical}(\theta_{\cdot|\boldsymbol{s}_t}).$$

The base distribution, $p_0$, is defined over strings in $\Sigma^* \cup \{\langle/\mathrm{S}\rangle\}$ by deciding with a specified probability to end the utterance, a geometric length model, and a uniform probability over $\Sigma$ at a each position. Intuitively, it captures the preference for having short words in the lexicon. In addition to the HDP model, we also evaluate a simpler single Dirichlet process (DP) version of the model, in which the $\boldsymbol{s}_t$'s are generated directly as draws from $\mathrm{Categorical}(\theta_{\cdot})$.

By integrating out the draws from the DP's, it is possible to do inference using Gibbs sampling directly in the space of segmentation decisions. We use 1,000 iterations with annealing to find an approximation of the MAP segmentation and then use the corresponding posterior predictive distribution to estimate the held-out likelihood assigned by the model, marginalizing the segmentations using appropriate dynamic programs. The evaluated segmentation was the most probable segmentation according to the posterior predictive distribution.

In the original Bayesian segmentation work, the hyperparameters (i.e., $\alpha_0$, $\alpha_1$, and the components of $p_0$) were selected subjectively. To make comparison with our neural models fairer, we instead used an empirical approach and set them using the held-out likelihood of the validation set. However, since this disadvantages the DP/HDP models in terms of segmentation, we also report the original results on the BR corpora.

**Deterministic Baselines**   Incremental word segmentation is inherently ambiguous (e.g., the letters *the* might be a single word, or they might be the beginning of the longer word *theater*). Nevertheless,

several deterministic functions of prefixes have been proposed in the literature as strategies for discovering rudimentary word-like units hypothesized for being useful for bootstrapping the lexical acquisition process or for improving a model's predictive accuracy. These range from surprisal criteria (Elman, 1990) to sophisticated language models that switch between models that capture intra- and inter-word dynamics based on deterministic functions of prefixes of characters (Chung et al., 2017; Shen et al., 2018).

In our experiments, we also include such deterministic segmentation results using (1) the surprisal criterion of Elman (1990) and (2) a two level hierarchical multiscale LSTM (Chung et al., 2017), which has been shown to predict boundaries in whitespace-containing character sequences at positions corresponding to word boundaries. As with all experiments in this paper, the BR-corpora for this experiment do not contain spaces.

**SNLM Model configurations and Evaluation**   LSTMs had 512 hidden units with parameters learned using the Adam update rule (Kingma & Ba, 2015). We evaluated our models with bits-per-character (bpc) and segmentation accuracy (Brent, 1999; Venkataraman, 2001; Goldwater et al., 2009). Refer to Appendices B–D for details of model configurations and evaluation metrics.

## 6   RESULTS

In this section, we first do a careful comparison of segmentation performance on the phonemic Brent corpus (BR-phono) across several different segmentation baselines, and we find that our model obtains competitive segmentation performance. Additionally, ablation experiments demonstrate that both lexical memory and the proposed expected length regularization are necessary for inferring good segmentations. We then show that also on other corpora, we likewise obtain segmentations better than baseline models. Finally, we also show that our model has superior performance, in terms of held-out perplexity, compared to a character-level LSTM language model. Thus, overall, our results show that we can obtain good segmentations on a variety of tasks, while still having very good language modeling performance.

**Word Segmentation (BR-phono)**   Table 1 summarizes the segmentation results on the widely used BR-phono corpus, comparing it to a variety of baselines. **Unigram DP**, **Bigram HDP**, **LSTM suprisal** and **HMLSTM** refer to the benchmark models explained in §5. The ablated versions of our model show that without the lexicon (−memory), without the expected length penalty (−length), and without either, our model fails to discover good segmentations. Furthermore, we draw attention to the difference in performance of the HDP and DP models when using subjective settings of the hyperparameters and the empirical settings (likelihood). Finally, the deterministic baselines are interesting in two ways. First, LSTM surprisal is a remarkably good heuristic for segmenting text (although we will see below that its performance is much less good on other datasets). Second, despite careful tuning, the HMLSTM of Chung et al. (2017) fails to discover good segments, although in their paper they show that when spaces are present between, HMLSTMs learn to switch between their internal models in response to them.

Furthermore, the priors used in the DP/HDP models were tuned to maximize the likelihood assigned to the validation set by the inferred posterior predictive distribution, in contrast to previous papers which either set them subjectively or inferred them Johnson & Goldwater (2009). For example, the DP and HDP model with subjective priors obtained 53.8 and 72.3 F1 scores, respectively (Goldwater et al., 2009). However, when the hyperaparameters are set to maximize held-out likelihood, this drops obtained 56.1 and 56.9. Another result on this dataset is the feature unigram model of Berg-Kirkpatrick et al. (2010), which obtains an 88.0 F1 score with hand-crafted features and by selecting the regularization strength to optimize segmentation performance. Once the features are removed, the model achieved a 71.5 F1 score when it is tuned on segmentation performance and only 11.5 when it is tuned on held-out likelihood.

**Word Segmentation (other corpora)**   Table 2 summarizes results on the BR-text (orthographic Brent corpus) and Chinese corpora. As in the previous section, all the models were trained to maximize held-out likelihood. Here we observe a similar pattern, with the SNLM outperforming the baseline models, despite the tasks being quite different from each other and from the BR-phono task.

|  | Precision | Recall | F1 |
|---|---|---|---|
| LSTM surprisal (Elman, 1990) | 54.5 | 55.5 | 55.0 |
| HMLSTM (Chung et al., 2017) | 8.1 | 13.3 | 10.1 |
| Unigram DP | 63.3 | 50.4 | 56.1 |
| Bigram HDP | 53.0 | 61.4 | 56.9 |
| SNLM ($-$memory, $-$length) | 54.3 | 34.9 | 42.5 |
| SNLM ($+$memory, $-$length) | 52.4 | 36.8 | 43.3 |
| SNLM ($-$memory, $+$length) | 57.6 | 43.4 | 49.5 |
| SNLM ($+$memory, $+$length) | **81.3** | **77.5** | **79.3** |

Table 1: Summary of segmentation performance on phoneme version of the Brent Corpus (**BR-phono**).

|  |  | Precision | Recall | F1 |
|---|---|---|---|---|
| BR-text | LSTM surprisal | 36.4 | 49.0 | 41.7 |
|  | Unigram DP | 64.9 | 55.7 | 60.0 |
|  | Bigram HDP | 52.5 | 63.1 | 57.3 |
|  | SNLM | **68.7** | **78.9** | **73.5** |
| PTB | LSTM surprisal | 27.3 | 36.5 | 31.2 |
|  | Unigram DP | 51.0 | 49.1 | 50.0 |
|  | Bigram HDP | 34.8 | 47.3 | 40.1 |
|  | SNLM | **54.1** | **60.1** | **56.9** |
| CTB | LSTM surprisal | 41.6 | 25.6 | 31.7 |
|  | Unigram DP | 61.8 | 49.6 | 55.0 |
|  | Bigram HDP | 67.3 | 67.7 | 67.5 |
|  | SNLM | **78.1** | **81.5** | **79.8** |
| PKU | LSTM surprisal | 38.1 | 23.0 | 28.7 |
|  | Unigram DP | 60.2 | 48.2 | 53.6 |
|  | Bigram HDP | 66.8 | 67.1 | 66.9 |
|  | SNLM | **75.0** | **71.2** | **73.1** |

Table 2: Summary of segmentation performance on other corpora.

**Word Segmentation Qualitative Analysis**   We show some representative examples of segmentations inferred by various models on the BR-text and PKU corpora in Table 3. As reported in Goldwater et al. (2009), we observe that the DP models tend to undersegment, keep long frequent sequences together (e.g., they failed to separate articles). HDPs do successfully prevent oversegmentation; however, we find that when trained optimize held-out likelihood, they often insert unnecessary boundaries between words, such as *yo u*. Our model's performance is better, but it likewise shows a tendency to oversegment. Interestingly, we can observe a tendency tends to put boundaries between morphemes in morphologically complex lexical items such as *dumpty 's*, and *go ing*. Since morphemes are the minimal units that carry meaning in language, this segmentation, while incorrect, is at least plausible. Turning to the Chinese examples, we see that both baseline models fail to discover basic words such as 山间 (mountain) and 人们 (human).

Finally, we observe that none of the models successfully segment dates or numbers containing multiple digits (all oversegment). Since number types tend to be rare, they are usually not in the lexicon, meaning our model (and the H/DP baselines) must generate them as character sequences.

**Language Modeling**   The above results show that the SNLM infers good word segmentations. We now turn to the question of how well it predicts held-out data. Table 4 summarizes the results of the language modeling experiments. Again, we see that SNLM outperforms the Bayesian models and a character LSTM. Although there are numerous extensions to LSTMs to improve language modeling performance, LSTMs remain a strong baseline (Melis et al., 2018).

One might object that because of the lexicon, the SNLM has many more parameters than the character-level LSTM baseline model. However, unlike parameters in LSTM recurrence which are used every

|  |  | Examples |
|---|---|---|
| **BR-text** | Reference | are you going to make him pretty this morning |
|  | Unigram DP | areyou goingto makehim pretty this morning |
|  | Bigram HDP | areyou go ingto make him p retty this mo rn ing |
|  | SNLM | are you go ing to make him pretty this morning |
|  | Reference | would you like to do humpty dumpty's button |
|  | Unigram DP | wouldyoul iketo do humpty dumpty 's button |
|  | Bigram HDP | would youlike to do humptyd umpty 's butt on |
|  | SNLM | would you like to do humpty dumpty 's button |
| **PKU** | Reference | 笑声 、 掌声 、 欢呼声 ， 在 山间 回荡 ， 勾 起 了 人们 对 往事 的 回忆 。 |
|  | Unigram DP | 笑声 、 掌声 、 欢呼 声 ， 在 山 间 回荡 ， 勾 起 了 人们对 往事 的 回忆 。 |
|  | Bigram HDP | 笑 声 、 掌声 、 欢 呼声 ， 在 山 间 回 荡 ， 勾 起 了 人 们对 往事 的 回忆 。 |
|  | SNLM | 笑声、 掌声 、 欢呼声 ， 在 山间 回荡 ， 勾起 了 人们 对 往事 的 回忆 。 |
|  | Reference | 不得 在 江河 电缆 保护区 内 抛锚 、 拖锚 、 炸鱼 、 挖沙 。 |
|  | Unigram DP | 不得 在 江河 电缆 保护 区内抛锚、 拖锚、 炸鱼、 挖沙 。 |
|  | Bigram HDP | 不得 在 江 河 电缆 保护 区内 抛 锚、 拖 锚 、 炸鱼、 挖沙 。 |
|  | SNLM | 不得 在 江河 电缆 保护区 内 抛锚 、 拖锚、 炸鱼 、 挖沙 。 |

Table 3: Examples of predicted segmentations on English and Chinese.

timestep, our memory parameters are accessed very sparsely. Furthermore, we observed that an LSTM with twice the hidden units did not improve the baseline with 512 hidden units on both phonemic and orthographic versions of Brent corpus but the lexicon could. This result suggests more hidden units are useful if the model does not have enough capacity to fit larger datasets, but that the memory structure adds other dynamics which are not captured by large recurrent networks.

|  | BR-text | BR-phono | PTB | CTB | PKU |
|---|---|---|---|---|---|
| DP | 2.328 | 2.933 | 2.245 | 6.162 | 6.879 |
| HDP | 1.963 | 2.553 | 1.798 | 5.399 | 6.424 |
| LSTM | 2.026 | 2.621 | 1.653 | 4.944 | 6.203 |
| **SNLM** | **1.943** | **2.536** | **1.560** | **4.836** | **5.883** |

Table 4: Test set language modeling performance (bpc).

# 7 RELATED WORK

Learning to discover and represent temporally extended structures in a sequence is a fundamental problem in many fields. For example in language processing, unsupervised learning of multiple levels of linguistic structures such as morphemes (Snyder & Barzilay, 2008), words (Goldwater et al., 2009) and phrases (Klein & Manning, 2001) have been investigated. Recently, speech recognition have benefited from techniques that enable the discovery of subword units (Chan et al., 2017; Wang et al., 2017); however, in this work the optimally discovered substrings look very unlike orthographic words. The model proposed by Wang et al. (2017) is essentially our model without a lexicon or the expected length regularization, i.e., (−memory, −length). Beyond language, temporal abstraction in sequential decision making processes has been investigated for a long time in reinforcement learning. Option discovery in hierarchical reinforcement learning is formalized similarly to the approach we take (using semi-Markov decision processes where we use semi-Markov generative models), and the motivation is the same: high level options/words have very different relationships to each other than primitive actions/characters (Sutton et al., 1999; Precup, 2000; Kulkarni et al., 2016).

# 8 CONCLUSION

We introduced the segmental neural language model which combines a lexicon and a character-level word generator to produce a model that both improves language modeling performance over word-agnostic character LSTMs, and it discovers latent words as well as the best existing approaches for unsupervised word discovering. This constellation of results suggests that structure discovery and predictive modeling need not be at odds with one another: the structures we observe in nature are worth modeling, even with powerful learners.

## A DATASET STATISTICS

Table.5 summarize dataset statistics.

| | Sentence | | | Char. Types | | | Word Types | | | Characters | | | Average Word Length | | |
|---|---|---|---|---|---|---|---|---|---|---|---|---|---|---|---|
| | Train | Valid | Test | Train | Valid | Test | Train | Valid | Test | Train | Valid | Test | Train | Valid | Test |
| BR-text | 7832 | 979 | 979 | 30 | 30 | 29 | 1237 | 473 | 475 | 129k | 16k | 16k | 3.82 | 4.06 | 3.83 |
| BR-phono | 7832 | 978 | 978 | 51 | 51 | 50 | 1183 | 457 | 462 | 104k | 13k | 13k | 2.86 | 2.97 | 2.83 |
| PTB | 42068 | 3370 | 3761 | 50 | 50 | 48 | 10000 | 6022 | 6049 | 5.1M | 400k | 450k | 4.44 | 4.37 | 4.41 |
| CTB | 50734 | 349 | 345 | 160 | 76 | 76 | 60095 | 1769 | 1810 | 3.1M | 18k | 22k | 4.84 | 5.07 | 5.14 |
| PKU | 17149 | 1841 | 1790 | 90 | 84 | 87 | 52539 | 13103 | 11665 | 2.6M | 247k | 241k | 4.93 | 4.94 | 4.85 |

Table 5: Summary of Dataset Statistics.

## B SNLM MODEL CONFIGURATION

For each RNN based model we used 512 dimensions for the character embeddings and the LSTMs have 512 hidden units. All the parameters, including character projection parameters, are randomly sampled from uniform distribution from $-0.08$ to $0.08$. The initial hidden and memory state of the LSTMs are initialized with zero. A dropout rate of 0.5 was used for all but the recurrent connections.

To restrict the size of memory, we stored substrings which appeared $F$-times in the training corpora and tuned $F$ with grid search. The maximum length of subsequences $L$ was tuned on the held-out likelihood using a grid search. Tab. 6 summarizes the parameters for each dataset. Note that we did not tune the hyperparameters on segmentation quality to ensure that the models are trained in a purely unsupervised manner assuming no reference segmentations are available.

| | max len (L) | min freq (F) | $\lambda$ |
|---|---|---|---|
| BR-text | 10 | 10 | 7.5e-4 |
| BR-phono | 10 | 10 | 9.5e-4 |
| PTB | 10 | 100 | 5.0e-5 |
| CTB | 5 | 25 | 1.0e-2 |
| PKU | 5 | 25 | 9.0e-3 |

Table 6: Hyperparameter values used.

## C LEARNING

The models were trained with the Adam update rule (Kingma & Ba, 2015) with a learning rate of 0.01. The learning rate is divided by 4 if there is no improvement on development data. The maximum norm of the gradients was clipped at 1.0.

## D EVALUATION METRICS

**Language Modeling** We evaluated our models with bits-per-character (bpc), a standard evaluation metric for character-level language models. Following the definition in Graves (2013), bits-per-character is the average value of $-\log_2 p(x_t \mid \boldsymbol{x}_{<t})$ over the whole test set,

$$bpc = -\frac{1}{|\boldsymbol{x}|} \log_2 p(\boldsymbol{x}),$$

where $|\boldsymbol{x}|$ is the length of the corpus in characters. The bpc is reported on the test set.

**Segmentation** We also evaluated segmentation quality in terms of precision, recall, and F1 of word tokens (Brent, 1999; Venkataraman, 2001; Goldwater et al., 2009). To get credit for a word, the

models must correctly identify both the left and right boundaries. For example, if there is a pair of a reference segmentation and a prediction,



Reference: `do you see a boy`

Prediction: `doyou see a boy`



then 4 words are discovered in the prediction where the reference has 5 words. 3 words in the prediction match with the reference. In this case, we report scores as precision = 75.0 (3/4), recall = 60.0 (3/5), and F1, the harmonic mean of precision and recall, 66.7 (2/3). To facilitate comparison with previous work, segmentation results are reported on the union of the training, validation, and test sets.

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
