# OpenReview forum: "Unsupervised Word Discovery with Segmental Neural Language Models"
_ICLR.cc/2019/Conference_

### Official Review · AnonReviewer3 · 2018-10-30
**Interesting research direction, novel contributions, and well-written paper**

**Rating:** 6
**Confidence:** 4

**Review:**

This paper presented a novel approach for modeling a sequence of characters as a sequence of latent segmentations. The challenge here was how to efficiently compute the marginal likelihood of a character sequence (exponential number different of segmentations). The author(s) overcame this by having a segment generation process independent from the previous segment (only depends on a sequence of characters). The inference is then required a forward algorithm. To generate a segment, a model can either select a lexical unit (pre-processed from a training corpus) or generate character by character.

On the experiments, the author(s) showed that the model recovered semantical segmentation on many word segmentation dataset (including phonemes). The lexical memory and the length regularization both contribute significantly as shown in the analysis. The language modeling result (BPC) was also competitive with LSTM-based LMs.

I think the overall model is interesting and well motivated, though it is a bit disappointing that the author(s) needed to use an extra regularizer to constraint the segment length (from the lexical memory?). Perhaps, the way they build a lexical memory should be investigated further. The experiment should also show an evidence that SNLM(+memory, -length) was overfitted as claimed.

The validation and test dataset have been modified to remove "samples" containing OOV characters. How many have been removed? The author(s) could opt for an unknown character similar to many word-level datasets.

The use of word segmentation data was quite clever, but this also downplayed other work that is not aimed to recover human-semantic segmentations. For example, a segment "doyou" on page 10 might be considered as a valid segmentation since it appears a whole lot. HM-LSTM though did poorly on the segmentation task but performed rather well on PTB LM task, but the author(s) decided to omit this comparison.

Some minor comments:
- A typo in the introduction "... semi-Markov model. The the characters inside ...".
- Eq 3 is a bit hard to follow. Perhaps, a short derivation should be presented.
- Is it possible to efficiently generate a sequence?

[Updated after reconsidering other reviews]
Although this paper misses some related work and comparison models, I think it still has a valid contribution to language modeling: a character-level language model that produces plausible word segmentation.

---

> ### Author Response · Authors · 2018-11-20
> **In response to AnonReviewer3**
>
> Thank you for the comments.
>
> Regarding the question about the regularizer. Every model we know of in the literature that contains a nonparametric lexicon has had to have some explicit mechanism for preventing degenerate solutions (variously based on MDL criteria, nonparametric priors, traditional regularization criteria); while it’s possible that the regularization due to drop-out or early stopping would be sufficient to prevent overfitting, those methods are tailored to the neural net aspect of the model (and used here), but we do not find it surprising that a model containing a new kind of memory would also need a new kind of capacity control. We see this not as a surprise or disappointment, but rather as another weak confirmation that the model is behaving as we expect.
>
> Regarding changes to the valid/test data, we removed a single utterance from the validation data (since it contained the rare phoneme ‘zh’ in the word pleasure, which was not present in the training data).
> Regarding the performance of models that discover “bad” segments but have good predictive distributions. There are indeed many such models, and traditional statistical criteria like mutual information discover many “non-words”. We will clarify this in the paper, but we wish to refer the reviewers and area chairs to our introduction which articulates that there is intrinsic scientific value in developing models that can chunk unsegmented input into actual words (as humans do).

---

### Official Review · AnonReviewer1 · 2018-11-03
**Very little awareness on latest research on unsupervised word segmentation**

**Rating:** 3
**Confidence:** 5

**Review:**

This paper proposes a neural architecture for segmental language modeling
that enables unsupervised word discoveries. The architecture employes a
two-stage architecture that a word might be a type, or a sequence of characters
of its spellings.
This idea is basically similar to Nested Pitman-Yor language models
(Mochihashi et al. 2009) and two-stage language models (Goldwater et al. 2011),
but the authors seem not to notice these previous work.
Experimental results show some improvements on naive baselines, but clearly
below the state-of-the-art in unsupervised word segmentation.

As noted above, the crucial drawback of this paper is that the authors are
completely unaware of latest achievements on unsupervised word segmentation
and discovery, rather than old, simplistic baselines such as Goldwater+ (2009,
idea is based on Goldwater+ ACL 2006) or Berg-Kirkpatrick (2010).
The idea of using characters and words is already exploited in Mochihashi+
(ACL 2009) in a nonparametric Bayesian framework; it has a better F1 than this
work by a large margin. Moreover, it is recently extended (Uchiumi+ TACL
2015) to also include latent word categories as well as segmentations to yield
the state-of-the-art accuracies on F1=81.6 on PKU corpus, as compared to 73.1
in this paper.
Note that they employ a prior distribution on segment lengths as a (mixture of)
Poisson distributions or negative binomials whose parameters are
automatically learned during inference, as compared to a post-hoc regularization
used in this paper.

In a Bayesian framework, interpolations between words and characters are
theoretically derived and quite carefully learned, and regularizations are
automatically adjusted. While neural architectures have some potentials
to improve over them, current heuristic architectures that have lower
performance does not have any advantage over these methods,
both theoretically and empicially.

---

> ### Author Response · Authors · 2018-11-06
> **In response to AnonReviewer1**
>
> This work presents a recurrent neural network language model that obtains better predictive distributions (perplexity) than an LSTM while also discovering the words that exist in language. The review above misconstrues the aim of this paper as simply producing the best segmentation accuracy, which the papers cited achieve by tuning for segmentation performance on held out data and sacrificing predictive accuracy. While we agree with the reviewer that there has been much excellent work done on nonparametric Bayesian segmentation (and two key ideas from that modeling tradition directly inspired this work!), no such model has been shown to achieve perplexities close to those of an RNN. However no previous RNN has been shown to discover plausible word segmentations. Our model achieves both. In doing so we argue that word segmentation is not a task that should be studied in isolation from the rest of the language learning but that the flexibility of neural models means they can approach other aspects (e.g., grounding in different modalities or tasks, learning large scale syntactic regularities) more naturally than would be practical with current Bayesian techniques.
>
> Below we elaborate on several more technical objections to this review:
>
> First, our decision to focus on DP/HDP models rather than the extensions referred to in the review (specifically PYP/HPYPs, nested PYPs, integrating out hyperparameters, etc.) was not due to ignorance, but rather that we were incorporating a two core ideas from Bayesian nonparametric word segmentation/language modeling into neural networks and we chose the simplest possible Bayesian model that made our points. These are: (1) that a lexicon that memorizes word chunks is useful for inducing good segmentations; (2) that capacity control is important when you have a lexicon like this. We do agree that nested PYPs, which learn to model character sequences (although not across word boundaries), deserve discussion and we will update the paper accordingly (again we emphasize that this is an oversight, not something that changes the meaningfulness of our results). Thus, the DP/HDP models otherwise perfectly illustrates the points they needed to illustrate, and the newer variations do not offer any additional insight into how to fix the problems that RNNs have with discovering words.
>
> Second, our results are not precisely comparable since Bayesian unsupervised learning has traditionally been evaluated in a setup which does not distinguish between a train and test set, or which uses observations from both when performing posterior inference. As we demonstrated, in Bayesian models, selecting hyperparameters empirically (ie, based on held-out likelihood) results in less effective structure discovery than setting the hyperparameters subjectively (however, since some standard datasets did not have a train/test split until this paper, we expect that in many cases, these models were chosen based on reported segmentation accuracies!). We certainly appreciate the insights that have been enabled by using both methodologies, but our perspective is that relying on held-out likelihood for model selection is eminently defensible methodology. However, held-out likelihood is indeed a radically different development/training/evaluation methodology for working on segmentation that is a better fit for neural models (and, we think, Bayesian models as well) than what came before, and it does make the results incomparable. Finally, another source of incompatibility is that the Uchiumi et al (2015) length distribution correction relies on hand-engineered features. We expected these were selected to improve reported segmentation accuracy (rather than validation likelihood), and as an ICLR paper, we are exploring how well we can do with learning representations, rather than engineering them.
>
> Third, the goal of this paper was to show that it is possible to align the goals of good segment discovery with good held-out models of language (after all, humans are good at both!). In their zeal to argue that our segmentation accuracy lags behind that of the best Bayesian models (which we questioned in the previous paragraph), the reviewers ignore the crucial fact that the most basic RNNs outperform the best hierarchical Bayesian language models by far in terms of predicting held-out data. Surprisingly, although posterior predictive checking is a standard tool for assessing Bayesian models, none of the existing Bayesian segmentation papers seem to have used this methodology, and so we had to include our own likelihood experiments (Table 4) to demonstrate this disparity. These results show clearly that while Bayesian models are perhaps slightly better than our models in terms of segmentation accuracy, they are far less good than RNNs in terms of predictive accuracy. On the other hand, our RNNs are good at both.

---

> > ### Comment · AnonReviewer1 · 2018-11-06
> > **Authors are still misunderstanding previous work**
> >
> > Reading the rebuttal, I found that the authors do not understand well what were already done in previous work, which performs far better than this paper.
> >
> > - 1st, "a lexicon that memorizes word chunk" is exactly what is already done by nested PYP or two-stage language models. In such nonparametric models, the lexicon is regarded as a base measure of language models and its atoms, i.e. words, are created when necessary and reused afterwords by a dynamic interpolation. This is exactly the core idea that this paper claims to be novel.
> >
> > - 2nd, Bayesian segmenting model also produces good predictive accuracy as a language model. Therefore, the argument that this work only aims to produce good predictive accuracy is invalid.
> >
> > - 3rd, in Bayesian models, "held-out likelihood" is unnecessary in general. It is in fact
> > a general advantage of Bayesian approaches that we do not need such held out data, but just maximizing the likelihood of training data would yield good generalization performance. In fact, "selecting hyperparameters empirically (ie, based on held-out likelihood) " in the rebuttal is not true; hyperparameters are just sampled from its posterior using training data only, and it produces very good accuracy. I think this misunderstanding came from the fact that the authors only use a frequentist perspective.
> >
> > - 4th, evaluations in previous work are just done by splitting training and test data.
> > Therefore, "Bayesian unsupervised learning has traditionally been evaluated in a
> > setup which does not distinguish between a train and test set" is simply false.
> > Once the model is learned from training data, the model is fixed and the
> > segmentation on test data is computed by a Viterbi decoding (but it should not
> > always be so because it is stochastic).
> >
> > I know very well that RNN language models have superior  performance than n-gram
> > language models, and it fits better to multimodal situations especially.
> > Therefore, lately many segmenting neural models are proposed, but almost
> > all of them fail to yield good segmentations that Bayesian approaches already
> > achieved because these Bayesian work have a structured prior constructed to fit to
> > natural language well, and the hyperparameters are tuned automatically to maximize
> > the performance of language models, as opposed to frequentist methods that require
> > held-out data or heuristic regularization parameter.
> > Therefore, in general I think that neural methods should incorporate these previous
> > findings in any way to combine its superior predictive performance to have a better
> > segmenting language models in the future.

---

> > > ### Author Response · Authors · 2018-11-20
> > > **In response to AnonReviewer1**
> > >
> > > We are puzzled by this unconstructive review. Having published numerous papers on non-parametric Bayesian models of language (including ones cited in the papers the reviewer suggests we do not understand) we hoped it was clear that this work is directly aiming to bring the advantages of Bayesian models of segmentation to state-of-the-art neural language models, while avoiding their disadvantages (e.g. there is nothing Bayesian or principled about Viterbi decoding from a point estimate derived from single Gibbs sample, we know as we have done it!).
> > > If there is a specific reference that the reviewer can point to that demonstrates a Bayesian segmentation model achieving perplexities close to the current state of the art we would be delighted to see such a results. Conversely, if there is a reference that demonstrates a neural model achieving unsupervised segmentation performance anywhere near what are reported in our paper we would also be keen to know this.

---

### Official Review · AnonReviewer4 · 2018-11-08
**Some interesting ideas, but comparison to previous work might be misleading**

**Rating:** 4
**Confidence:** 3

**Review:**

[Note to the authors: I was assigned this paper after the reviewing deadline.]

The authors train language models on unsegmented text, simultaneously discovering word boundaries
without direct supervision. Given the past history, but ignoring past segmentation decisions
to keep computations tractable, the model predicts the next character segment (word-like unit)
by combining a character-level LSTM with a lexical memory. To prevent overusing the
lexical memory, which would lead to poor generalization, the authors propose a segment
length penalty.

Strengths:

The model architecture is interesting, combining the benefits of a character-level
model (open vocabulary) with those of a lexical model (effective for frequent character
sequences).

Despite the exponential number of possible segmentations, inference remains tractable
using dynamic programming (with some simplifying assumptions).

The ablation study clearly shows that both the lexical memory and the length penalty
contribute significantly.

Weaknesses:

The writing quality is somewhat weak. Many errors should have been caught when
proofreading the paper (e.g. "The segmentation decisions and decisions" and
"The the characters" on page 1).

I am confused by the key-value pairs of the lexical memory. Shouldn't character
sequences be keys, and their trainable vector representations be values?

It is hard to evaluate how good the language models are, as the strength of the
baselines is unclear. How well-tuned is the LSTM?

Comparison to some other segmentation approaches (not necessarily with language modeling)
is limited. In particular, adaptor grammars perform very well on the Brent corpus [1].
However, [2] is mentioned briefly. As these other approaches work better for segmentation,
the authors should carefully justify why having a single model that does both language
modeling and word segmentation well matters. Many neural approaches have also been
suggested for Chinese word segmentation (among others [3]). In these papers, results on the
PKU dataset are much better.  Are these directly comparable with yours?

I would have liked a finer analysis of the impact of the length penalty.
A plot showing how validation likelihood and segmentation performance vary as
\lambda is increased could potentially be interesting.

[1] Johnson and Goldwater. "Improving nonparameteric Bayesian inference: experiments on
unsupervised word segmentation with adaptor grammars", HLT, 2009

[2] Berg-Kirkpatrick et al. "Painless Unsupervised Learning with Features", NAACL, 2010

[3] Yang et al. Yang, Jie, Yue Zhang, and Fei Dong. "Neural Word Segmentation with Rich Pretraining", ACL, 2017

---

> ### Author Response · Authors · 2018-11-20
> **In response to AnonReviewer4**
>
> Thank you very much for the review.
> The paper about Chinese segmentation ([3]) is a supervised segmentation model. Which is not comparable with our paper. As we discuss in the paper and in the reply to a comment posted below, the best performing segmentation models [2] rely on hyper-parameters tuned by looking at segmentation performance. However, we argue this is not purely unsupervised learning as it requires gold segmentations at training time. Those models will not work on zero-resource languages or multiple languages without manually tuning the parameters for each language. Nor does supervised heldout tuning provide a plausible model for human language acquisition. That’s why we think it’s important that our model, which is tuned to maximize unsupervised held-out likelihood, achieves competitive segmentation performance.
>
> The keys are vectors and values are string as we use vectors to query what strings to be generated.

---

### Public Comment · (anonymous) · 2018-10-09
**You can compare Segmental Language Model (SLM) with your SNLM**

You can mention [1] and its experimental results. SLM uses the same idea as SNLM and gives higher F1 (80.2) on PKU (while it has not been evaluated on Brent, PTB or CTB).

[1] Unsupervised Neural Word Segmentation for Chinese via Segmental Language Modeling. EMNLP 2018. [ https://arxiv.org/abs/1810.03167 ]

---

> ### Author Response · Authors · 2018-10-16
> **Author response to questions.**
>
> We will cite the paper and carefully discuss their results. Briefly, there are several differences that explain the difference. One major difference is the training scheme. If you look at the code for the EMNLP paper (https: //github.com/Edward-Sun/SLM), the model is being tuned to maximize the segmentation performance by running segmentation evaluation during training. We argue that this is not a purely unsupervised model as it requires gold segmentation data at training time (e.g., setting maxlen=2 is an extremely strong form of supervision), and in our work, we are crucially interested in a model whose word predictive likelihood correlates with (unobserved) segmentation performance.
>
> There are other differences in experimental setup.
> 1) The EMNLP paper uses pre-trained word embeddings trained on a large corpus.
> 2) The EMNLP paper preprocessed arabic numerals, punctuation symbols, and English words.

---

### Public Comment · (anonymous) · 2018-10-09
**Another possibly relevant work**

Another possibly relevant work that came out a couple of years ago is "Multiscale sequence modeling with a learned dictionary"[1]  might be of interest to you.

[1] Multiscale sequence modeling with a learned dictionary, https://arxiv.org/abs/1707.00762, MLSLP 2017.

---

> ### Author Response · Authors · 2018-10-16
> **Author response to questions.**
>
> Thank you for drawing our attention to this work, which we had managed to miss. It uses a similar strategy to populate its dictionary, but doesn’t include a character-based method for generating “inside words” (see the response above to C1 for our empirical findings regarding the importance of this component to segmentation quality).

---

> > ### Public Comment · (anonymous) · 2018-10-18
> > **Yes! Ofcourse**
> >
> > Yes ofcourse, this is not the same as the above mentioned work! I just wanted to draw your attention to the paper in case you had missed it.

---

### Public Comment · ~Sabrina_J_Mielke1 · 2018-10-09
**Some curious questions**

First of all, thank you for citing our paper on the usage of words and subwords in open-vocabulary language modeling ( https://arxiv.org/abs/1804.08205 )!
We are excited to see more work on the fascinating question of how to model linguistic sequences.
And speaking of questions, I have a few that I would appreciate a response to:

1) The abstract promises usage of the "structure learning mechanism of Bayesian non-parametrics", but that doesn't seem to be what is happening in the paper: the model is non-parametric in that the number of parameters is growing with the size of the training data (because "all" possible substrings from the training set are identified before training and used in a big lookup table), but the mechanisms of *Bayesian* non-parametrics, as they are used in previous work, leading up to the sequence memoizer (Wood et al., 2011 and earlier work) seem to be missing from this work. Is the abstract out of sync with the paper or am I missing where the promise of Bayesian non-parametrics is fulfilled?

2) The lookup table is also interesting for practical reasons. Appendix B mentions that only sequences of up to some tuned maximum length and minimum frequency are kept in this table: how big then does that table become in practice and how is a softmax over all contained elements made tractable?

3) Why include a model p_char in p(s)? Should the substrings not be enough once you relax the requirement of segments having at least length 2? Or does this lead to degenerate solutions in some sense?

4) What makes the comparison to deterministic segmentations like BPE or sentencepiece hard or impossible? Can one tune their respective parameters to perform well as language models (as you correctly cite, we know the answer is yes) and could we maybe even achieve decent enough segmentations (maybe even outperforming the surprisal criterion or at least the surprisingly bad HMLSTM)?
Also, as a note: it might be worth explaining, perhaps in another appendix, just how you obtain segmentations from Elman's surprisal and the HMLSTM -- right now, section 5 is just stressing that they are prefix-only (good point!) and "deterministic" in some sense (though we should easily be able to "sample" these, too, right?).

5) What necessitates the removal of spaces for the English data? In the interest of allowing comparison to previous work, why not keep them and observe whether the SNLM pastes spaces on the back or start of words? Does it do something uniform at all or does performance actually degrade?

6) On your point of number of parameters, is it fair to equate the memory bank with the number of hidden units in your baseline LSTM or should it maybe rather be compared to a multi-layer LSTM (assuming that all your models are single-layer), which we know are significantly more powerful than single-layer LSTM?

7) More open-ended: why is evaluating against the gold standard of your test sets meaningful? How are these gold standards defined and what makes them the "best" possible segmentation? Using, for example, spaces or the boundaries of existing tokenizers as gold bounds makes sense, but cross-linguistic analysis shows just how brittle and sometimes counterproductive the reliance on spaces is -- modeling morphemes may make more sense for morphologically rich languages. In fact, I am now wondering what would happen for less isolating languages, if you reproduced the samples from Table 3 (which I assume are already cherry-picked) for, say, Turkish?

I understand that the ICLR layout leaves little space to answer all these questions, but I would appreciate hearing your opinion on them.

---

> ### Author Response · Authors · 2018-10-16
> **Author response to questions.**
>
> Thank you very much for the interesting questions and giving us an opportunity to respond to questions that many readers might have had.
>
> Q1: The connection to Bayesian nonparametrics is just an analogy, referring to the fact that there is both a memorized component that grows unboundedly with the data and a penalty that prevents that excess capacity from being overused.
>
> Q2: We will add information about the softmax size to the appendix; it is big, around 20k, but it remains tractable.
>
> Q3: The character-level model enables the model to generate chunks that are not stored in memory. For example, numbers will not generally be in memory, but they should be a chunk. Empirically, we found removing p_char and putting single-length characters in the lexicon resulted in substantially worse results.
>
> Q4: BPE is a kind of unigram-model-based compression with capacity control provided by the limits on vocabulary size, and a particular greedy inference procedure -- and it can indeed be compared. The best segmentation result that would be possible with BPE (i.e., tuning the vocabulary size on segmentation accuracy) is P 24.97 R 36.59 F 29.68 which is better than the HMLSTM baseline but worse than LSTM surprisal baseline P 54.5 R 55.5 F 55.0. The optimal segmentation selected by likelihood would, presumably, be worse.
>
> As for the Elman’s surprisal baseline, we will clarify how we create these to make the paper more self-contained. For now, please refer to the original paper [https://crl.ucsd.edu/~elman/Papers/fsit.pdf]. Since they deterministically rely on changes in conditional entropy, rather than stochastic decisions, it’s not obvious how to sample from them.
>
> Q5: The focus of this paper is on unsupervised word discovery, which we think is not a very coherent problem when spaces that mark word boundaries are present. Although some recent prior work has done segmentation on data that includes spaces, the very long history of this problem has almost entirely been explored without spaces.
>
> Q6: The comparison is primarily to draw attention to the fact that the memory is accessed sparsely, parameter counts of the +mem model cannot be compared straightforwardly to a model without memory, and therefore that the improvements in perplexity aren’t just an artifact of “more parameters” (we also find that doubling the size of the character-only LSTMs did not result in improved performance). We will clarify this.
>
> Q7: Our view is that when it comes to intrinsic evaluation of segmentation, there are several possible “best answers” (e.g., any answer to the following questions can be justified: should bound morphemes be split? Should unbound morphemes be split? Should phonological words be the true units? Should idiosyncratic multiword expressions be the units? Should multiword expressions with transparent meanings be the units?); however, there are vastly many more truly bad segmentations (two bad segmentation of bad are “b +ad” and “ba +d”). Therefore, a more ideal evaluation should give credit for getting any reasonable segmentation, but only penalizing you for proposing a completely wrong segment/word (this is similar to the approach taken by Alignment Error Rate (Och & Ney, 2000) in assessing the quality of bilingual word alignments which supports both “sure” and “possible” alignments; and it is reminiscent of the approach taken by Dyer (2009), who trained a supervised word segmentation model to update toward a lattice of possible segmentations). Since there are no annotations of “possible segmentations” for the languages we were studying, we used conventional tasks and segmentation schemes for this problem to approximate this (although we do compare with two different Chinese segmentation standards). Regarding the question about Turkish, we expect that it would discover subword units since we already see evidence that morphemes like “ed” “ing” and “s” are segmented off in English in the likelihood-maximizing configuration.

---

> > ### Public Comment · ~Sabrina_J_Mielke1 · 2018-10-24
> > **Thanks!**
> >
> > Thanks for your detailed responses!
> > I think that many of these clarifications and additional details are very interesting and possibly of interest to other readers as well, so I hope you will be able to incorporate them into camera-ready or the appendix.
> >
> > Some small follow-ups:
> > For Q1, I now better understand what you meant, but it might be worth considering to drop or additionally qualify the word "Bayesian" -- it really did confuse me at first.
> > For Q2, I guess it is the frequency filtering that is key to getting, say, the PTB vocab from 6M (all length-10 strings) to 30k (only those that appear more than 100 times). I'd be curious how much the empirical success of the SNLM depends on the hyperparameter F (and, to a letter extent L and \lambda), but I understand that there is little space (and maybe reason) to report much on the grid search you did over both.
> >
> > Again, thanks for your detailed comments, I appreciate them.

---

### Meta-Review · Area_Chair1 · 2018-12-14
**reject**

**Confidence:** 4
**Recommendation:** Reject

**Metareview:**

a major issue or complaint from the reviewers seems to come from perhaps a wrong framing of this submission. i believe the framing of this work should have been a better language model (or translation model) with word discovery as an awesome side effect, which i carefully guess would've been a perfectly good story assuming that the perplexity result in Table 4 translates to text with blank spaces left in (it is not possible tell whether this is the case from the text alone.) even discounting R1, who i disagree with on quite a few points, the other reviewers also did not see much of the merit of this work, again probably due to the framing issue above.

i highly encourage the authors to change the framing, evaluate it as a usual sequence model on various benchmarks and resubmit it to another venue.